# Heat Not Burn Tobacco Product—A New Global Trend: Impact of Heat-Not-Burn Tobacco Products on Public Health, a Systematic Review

**DOI:** 10.3390/ijerph17020409

**Published:** 2020-01-08

**Authors:** Aleksandra Ratajczak, Piotr Jankowski, Piotr Strus, Wojciech Feleszko

**Affiliations:** 1Department of Pediatric Respiratory Diseases and Allergy, Medical University of Warsaw, Żwirki i Wigury 63A, PL-02-091 Warsaw, Poland; olarurarz@gmail.com (A.R.); struspiotrek@gmail.com (P.S.); 2Department of Internal Medicine, Pulmonary Diseases and Allergy, Medical University of Warsaw, PL-02-091 Warsaw, Poland; jankowskipiotr91@gmail.com

**Keywords:** heated tobacco products, heat-not-burn, IQOS, reduced risk, tobacco market, smoking, harm reduction

## Abstract

Introduction: The use of heat-not-burn tobacco products (HnB) is being adopted increasingly as an alternative to smoking combusted products, primarily cigarettes. Substantial controversy has accompanied their marketing and use in the public health context. In this study, we aimed to consider the probable impacts of HnB tobacco products use on public health. Methods: In May 2019, we conducted a systematic review of 15 studies concerning awareness and use of IQOS (abbrv. I Quit Ordinary Smoking) selected from three databases: Cochrane, PubMed, and Embase regarding public health. Results: All key outcomes varied by smoking status: more young adults who were currently smoking reported being aware of, interested in trying, and prone to trying heat-not-burn tobacco products. Interest in trying HnB products was also present among non-smokers, which raises concerns regarding new smokers. Interestingly, susceptibility to trying IQOS (25.1%) was higher than for traditional cigarettes (19.3%), but lower than for e-cigarettes (29.1%). Conclusions: Present studies suggest that HnB tobacco products have the potential to be a reduced risk product for public health compared to conventional cigarettes, considering indirectly the potential effects on the chronic diseases which are traditionally linked to traditional cigarette use as well as second hand exposure, but further studies are needed to determine whether this potential is likely to be realized. The process of HnB tobacco products becoming increasingly popular is of a global scale. Only small differences between countries on different continents regarding popularity and use of HnB tobacco products have been reported.

## 1. Introduction

The U.S. Federal Drug Administration (FDA) has approved the IQOS tobacco products intended for sale in the USA. Nonetheless, they still remain controversial in the context of health effects, due to the availability of only short-term observations. Recently, in 2019 a case report regarding a case of fatal acute eosinophilic pneumonia (AEP)—a rare disorder characterized by hypoxemia, pulmonary infiltrates, and pulmonary eosinophilia—related to the use of heat-not-burn (HnB) tobacco products was published [1]. The authors of the case reported a 16-year-old male who was admitted to an emergency department and intubated due to respiratory failure two weeks after smoking HnB. However, the health risks correlated to HnB tobacco products have not yet been clarified. It is important to note that according to the available literature, tobacco smoking is one of the main causes of AEP, however, it occurs secondary to drug exposure or hypersensitivity reactions to an inhaled antigen [2]. In this systematic review, we aimed to assess the impact of heat-not-burn tobacco products on public health.

Smoking cigarettes contributes to the development of numerous diseases, including asthma, cardiovascular disease (CVD), lung cancer, and COPD (chronic obstructive pulmonary disease) [3]. Due to the constantly underlined traditional cigarette smoking health risks, major tobacco companies have recently developed and began advertising alternative nicotine products, such as electronic cigarettes (e-cigarettes) and “heat-not-burn” cigarettes, which are tobacco companies’ latest products that do not burn tobacco. The first heat-not-burn tobacco product (HnB) called Premier™ (R.J. Reynolds) was introduced in 1988, followed by Eclipse™ (R.J. Reynolds) and Accord™ (Philip Morris) in the 1990s, Heat Bar™ (Philip Morris) in 2007, and iQOS™ (Philip Morris International) which was released in 2014 in Japan, Italy, and Switzerland [4]. These products heat tobacco (up to 350 °C) to generate an inhaled nicotine aerosol instead of burning it at significantly higher temperatures (around 800 °C) [5]. Although HnB tobacco products differ slightly, all of them seem to be a safer alternative to combustible cigarettes, according to several studies [6,7].

Heated tobacco product from Philip Morris International (PMI), IQOS (abbrev. I Quit Ordinary Smoking), consists of a charger, a holder and tobacco sticks, and plugs or capsules. A tobacco stick is inserted into the holder and the tobacco is heated with an electronically controlled heating blade which is inserted into the tobacco plug [8]. They are advertised as potentially reduced-harm products because, as the producer claims, their technology limits combustion and the generation of toxic compounds [9]. The principal argument for developing and marketing HnB is the declaration that they are considerably less dangerous than traditional cigarettes [10]. According to the study by St. Helen et al., which compares the amount of toxic compounds in the main stroke of traditional cigarettes with HnB aerosol, it seems justified. [11] Nevertheless, it is only one out of the few available independent studies to have verified the level of risk of these HnB, which raises an important debate for public health. A recent systematic review by Simonavicius et al. identified only three papers that addressed problems related to the public health issue among 31 identified studies [8]. This may be at least in part due to the lack of independent (non-tobacco-industry sponsored) evidence, since most of the available results come from tobacco industry data.

Therefore, in our study article we aimed to reliably analyze the impact of heat-not-burn tobacco products on public health.

## 2. Methods

The PRISMA (abbrv. preferred reporting items for systematic reviews and meta-analyses) was employed to guide this review.

### 2.1. Search Strategy

A systematic search of the literature was carried out to identify relevant studies published in English since 2014 to the 21st of May 2019. The following databases were used: PubMed, EMBase, and the Cochrane Central Register of Controlled Trials (CENTRAL). The keywords for our search included a combination of terms related to a potential effect of using IQOS (vascular endothelial function, cytotoxic effects, respiratory system, airway cells, lung function, brain, bronchial epithelial cells, toxicity, airway cells homeostasis, in vivo biomarkers, ultrafine particle pollution, second hand smoke exposure, COPD (chronic obstructive pulmonary disease), passive exposure, passive smoking, carcinogens, carcinogenic, cancer, oxidative stress, cytotoxic effects, toxic effects, cytotoxicity, respiratory health impairment, public health, animals, animal models, pollution, asthma, stroke, chemicals, smoke exposure, puff duration, cardiovascular effects) and terms related to HnB tobacco devices (heated tobacco products, heat-not-burn tobacco, cigarettes, heated tobacco products electronic nicotine delivery devices, IQOSTM, non-cigarette tobacco products, HTP, harm-reduction product, smoke-free electronic device with heatsticks, smoking substitution, nicotine delivery devices, HNB, reduced-risk tobacco product, non-combustion devices, smoking electronic device, e-cigarettes, glo, ploom).

The reference lists of the selected articles were subjected to a hand search to identify additional articles.

### 2.2. Selection Criteria

Studies were eligible for inclusion if they meet the Population, Intervention, Comparison, Outcome, and Study design (PICOS) criteria, namely:(1)Population: Heat-not-Burn tobacco product users or potential users;(2)Intervention: all types of interventions that are aimed at helping to understand and improve knowledge and awareness of HnB tobacco devices and their potential health effects among users or potential users;(3)Comparison: n/a;(4)Outcome: n/a;(5)Study design: survey studies.

Studies were excluded if they were: (1) not published in English; (2) not original paperwork.

### 2.3. Study Selection

Four investigators independently assessed and identified relevant articles to be included in this review. Studies were screened by title and abstract. Irrelevant or duplicate articles were excluded, and all remaining articles were subjected to full-text screening. Differences between the reviewers in the inclusion of articles were resolved through discussion and consensus. Figure 1 depicts the process of screening and including articles, and lists the reasons for excluding articles. Furthermore, in Table 1 studies (with their descriptions) included in the review were listed.

### 2.4. Data Extraction

Two authors extracted the following information from the included studies:(1)authors and years of publication;(2)city/area where the study was conducted;(3)funder;(4)study design;(5)main objective;(6)results;(7)number of participants

To sum up, 14 studies were taken into consideration. The whole process was pictured by the PRISMA flow diagram (Figure 1).

## 3. Public Health Issues

Findings from a study by Caputi et al. who analyzed data obtained by a Google search have revealed that HnB tobacco products will likely earn significant interest as they are introduced into new markets [12]. To prove this point, other research conducted in Canada, England, and the USA demonstrated that awareness of HnB products is expanding among adolescents [13]. In the study by Czoli et al., they analyzed youth awareness, interest in trying, and susceptibility to trying IQOS. Their results showed that awareness and interest in trying these products were very high among smokers, 7.0% and 38.6%, respectively. Interest in trying HnB products was also revealed among non-smokers. Susceptibility to trying IQOS (25.1%) was higher than for traditional cigarettes (19.3%) but lower than for e-cigarettes (29.1%). In another study conducted in 2018, Hair and colleagues highlight that as IQOS are marketed as a clean, chic, and pure product, they are mostly aimed at young adults [14]. This fact was observed demonstrated by new smokers who started using HnB products because of social reasons. Marynak et al. who conducted a study in 2017 revealed that circa one in 20 of U.S. adults were aware of HnB tobacco products, including one in ten who were current traditional cigarette smokers [15]. Overall, 0.7% of U.S. adults, including 2.7% of current smokers, reported use of HnB products. The usage was higher among current smokers than those who had never smoked, and higher among adults aged <30 years.

A slightly higher awareness of HnB products was reported by researchers from Georgia State University in a similar study performed in 2017 [16]. According to the study’s results, about one in eight adult study participants had heard of HnB tobacco products, 2.2% had used them, and 1.1% reported current use of HnB. The study also compared awareness, past use, and current use among all adults between 2016 and 2017—all increased significantly. The proportion of current use had more than doubled since 2016. Not surprisingly, as HnB gained popularity among younger people looking for innovations, adults under 45 years had higher rates of awareness than older adults. What cannot be explained so easily is an interesting fact that minority (non-white) participants had higher odds of past and current use of HnB compared with white participants.

A study investigating awareness and use of heat-not-burn products was also conducted in Europe, Great Britain [17]. According to its results, about 9% of adult citizens reported being aware of heat-not-burn products in a national survey. Less than 2% of respondents had tried them.

Japan, together with Italy and Switzerland, was one of the first countries where IQOS, one of the most recognizable HnB tobacco products, was introduced in 2014. Probably for this reason five out of 19 obtained studies concerning HnB products’ impact on public health were conducted in Japan.

As reported in the studies performed in the U.S., a significant increase in estimated rates of the current use of HnB tobacco products was also observed in Japanese adults [18]. In the years 2015 and 2016, only about 0.3% of respondents were current HnB users, whereas in 2017 its current user rate had increased more than 10 times and reached 3.6% of all respondents. Interestingly, the rapid increase in interest in and use of HnB products among Japanese was probably triggered by its appearance on a popular national entertainment TV show. The influential role of mass media’s impact on tobacco consumption trends could be observed following the show: among its viewers, HnB use was nearly four times higher than non-viewers (10.3% vs. 2.7%). The same study investigated health outcomes after secondhand HnB tobacco aerosol exposure. Nearly 12% of all respondents reported symptoms experienced after exposure to secondhand HnB tobacco aerosol. In that group, 37% experienced at least one symptom as a result, the most common being generally feeling ill (25.1%), followed by eye discomfort (22.3%), and sore throat (20.6%). Among current users of HnB products, the percentage of experienced symptoms was much lower (26%) in comparison with 41% of former users and 49% of people who had never used any tobacco product. The study also confirmed results from previous papers [19], showing that the younger population was more likely to use HnB products.

Another Japanese study examined electronic and HnB tobacco product use among 4432 chronic disease patients aged 40–69 years [20]. The percentage of heat-not-burn tobacco current or past use was low (<0.1%) among both men and women. These findings prove that HnB products do not arouse sufficient interest among the middle-aged group.

Lee et al. examined the association of HnB products use with perceived stress, physical activity, and internet use [21]. There was a negative association between HnB tobacco product use and perceived stress.

A study performed in 2015 with over seven thousand Japanese adults [22] found no associations between education background and past heat-not-burn tobacco use.

However, contradictory results were reported in a study conducted in Hong Kong in 2017 and published in 2019 [23]. According to the study, higher socioeconomic status (higher educational attainment and monthly household income) was associated with HnB use and its intention to use. The authors of the study underline the need for intensive public health education about HnB tobacco products, particularly for this high-risk group. Interestingly, only 11.3% of examined Hong Kong citizens were aware of HnB tobacco products and 1.0% had previously used it—probably due to the fact that HnBs were not yet legally approved on the market in that country.

A Korean study showed that all of the current IQOS users who were participants in their study were triple users of HnB tobacco products, conventional cigarettes, and e-cigarettes [24]. This finding contradicted the tobacco industry′s claims that traditional cigarette smokers will switch to HnB tobacco products. Lee et al. compared the estimated population health impact of introducing a Reduced Risk (tobacco) Product (RRP) into Japan under alternative assumptions about its rate of uptake and performed, for comparative purposes, a similar set of estimates for the USA [25]. They used data from the IQOS level of uptake as RRP use. According to this study, introducing the RRP (which was synonymous with IQOS in this research) substantially reduced smoking-related deaths. In Japan, the reduction over 20 years of four major diseases linked to traditional cigarette smoking (i.e., lung cancer, ischemic heart disease, stroke, and chronic obstructive pulmonary disease) will range from 8.5% to 11.4% in males and 13.7–17.7% in females, as estimated. Estimated results for the USA patients were lower because IQOS uptake rates are lower.

Levy et al. compared patterns of vaporized nicotine products (VNP) and traditional cigarette use in determining health effects [26]. In the group of VPN, the authors included e-cigarettes, pressurized aerosol nicotine products, and HnB tobacco products. The results implicated a strong potential for VNP use to improve population health by reducing or displacing cigarette use in countries where cigarette prevalence is high and smokers are interested in quitting. In their study, they underlined that current smokers were at least 15 times more likely to use VNPs than people who had never smoked.

## 4. Discussion

### 4.1. Principal Findings

First of all, to our knowledge this is the first study to show the global HnB range. Our systematic review shows that there are several main issues in the HnB tobacco products in the area of public health. Firstly, despite cultural differences in different continents, similar results were obtained regarding the popularity of HnB in many countries around the world. Secondly, HnB tobacco products are becoming more and more popular among young adults, even in non-smoking groups. Of note, the interest in HnB in the patient group aged 40–69 was much lower. There is a significant impact on advertising, which presents HnB as safe and clean, on HnB tobacco device usage. Another incentive to try this kind of device is a documented fact that HnB tobacco products seem to reduce smoking-related deaths [27]. What is more, studies revealed that higher socioeconomic status is associated with HnB usage.

### 4.2. Methodology

To obtain a good-quality study and reliable evidence, a precise methodology has to be done. Hence, finding as many studies related to HnB tobacco products′ impact on public health as possible was our general approach. We believe we met our expectations. Nevertheless, after rechecking we found one more study [1] which was added to our systematic review. Secondly, we aimed to make our study clear and easy to read. To achieve this, we used a helpful and clarifying PRISMA 2009 flow diagram added as a methodology figure. Furthermore, we rejected a significant number of studies (over 23,000). It is well-known that a proper review needs to be double-checked. Therefore, all of the authors performed the rejection process twice separately to be sure that none of the appealing studies has been missed.

### 4.3. Strong Points of the Study

To our knowledge, the current study is the first systematic review to examine HnB tobacco products’ impact on public health, including awareness and interest in trying HnB, in national samples across several countries. It needs to be pointed out that the tobacco industry sponsored the vast majority of studies examining the influence of HnB on human health. On the other hand, 14 out of 15 papers we found were conducted by independent research teams. Only one study was affiliated with a tobacco manufacturer.

### 4.4. Potential Confounds and Limitations of the Study

This systematic review is not without limitations. Firstly, the majority of identified studies were based on online surveys. Hence, their results may be different from those from a population-based study. Secondly, the studies we identified did not cover the potential impact of all HnB tobacco products on public health due to their variety of international markets and due to the novelty of those products. Also, the majority of the identified studies considered only IQOS. Although it appears to be the most prominent of the new generation of HnB tobacco products, products like Glo or Ploom earned considerate parts of the tobacco market in several countries. Some studies measured the HnB awareness by using images and descriptions of only some types and brands of HnB. Some of these products are no longer available. It is possible that some study participants responders might not have identified some of the products, either because they did not see their particular brand or did not recognize the product they used to fit within the definition. It is also probable that some respondents confused the product either with a conventional cigarette or an e-cigarette.

### 4.5. Comparison with Other Studies

Our review provides the first comprehensive summary of up-to-date evidence on HnB tobacco products’ impact on public health. This systematic review included 15 studies on HnB tobacco product’s public health impact of which 14 were independent. To compare, in this review Simonavicius et al. [5] identified 31 studies. However, 20 out of 31 of them were affiliated with the tobacco industry. Moreover, only three identified by Simonavicius were dedicated to the topic of health impact. The authors underlined in their study the lack of independent evidence. Most of the results used in their article were primarily drawn from tobacco industry data. The main objective of their study was to review peer-reviewed evidence on HnB, their secondhand emissions, and use by humans, whereas we concentrated on the public health implications of the introduction of heat-not-burn products on the market. There are some chemical and/or toxicological data relevant to differences in adverse health effects among the IQOS and other HnB tobacco products. A study published in 2018 has showed that short-term use of a specific HnB product, IQOS, may be effective to momentarily reduce acute cigarette craving while having a minimal impact on the eCO (abbrev. exhaled carbon monoxide) levels [28]. Another study from 2017 has concentrated on the hazards and toxicity of iQOS. The authors evaluated several harmful compounds (nicotine, tar, carbon monoxide (CO) and tobacco-specific nitrosamines (TSNAs)) in the mainstream smoke and fillers of iQOS and compared their concentrations with those from conventional combustion cigarettes. The concentrations of nicotine in tobacco fillers and the mainstream smoke of iQOS were almost the same as those of conventional combustion cigarettes, while the concentration of TSNAs was one fifth and CO was one hundredth of those of conventional combustion cigarettes. Although the results look promising, it is important to acknowledge that the toxic compounds are not completely removed from the mainstream smoke of iQOS, making it necessary to consider the health effects and regulation of iQOS in further research [29]. A study published recently in 2019 has pointed out that one of the HnB tobacco products’ manufacturers has applied for a classification as a Modified Risk Tobacco Product (MRTP) in the United States based on toxicological studies that have been published. However, it was underlined that data are not yet sufficient for a reliable assessment or recognition of potentially reduced health risks [30], which is consistent with our point of view described in this manuscript.

## 5. Conclusions and Further Research

Present studies suggest that HnB tobacco products have the potential to be a reduced risk product for public health compared to traditional cigarettes, especially when indirectly considering the potential effects on the chronic diseases which are traditionally linked to traditional cigarette use (i.e., CVDs, cancer, COPD) [25,31]. Nonetheless, further studies are needed to determine whether this potential is likely to be realized. Moreover, since HnBs are relatively new products (they have been on the market for five years), the long-term effects are still unknown. Although they are new on the market, they are gaining more and more popularity and have already almost reached the level of e-cigarette popularity in society. Nevertheless, they remain slightly less popular compared to e-cigarettes. The process of HnB tobacco products becoming increasingly popular is on a global scale. Only small differences between countries on different continents regarding the popularity and use of HnB tobacco products have been reported.

It is worth mentioning that current studies demonstrate clearly that HnBs are products on the rise. Their target group is young adults, which is a group with the most substantial economic prospects. HnBs are perceived as a prestigious gadget, so they appeal to people with a higher socio-economical level. It seems that people are convinced of the product’s better environmental performance compared to traditional cigarettes because it does not emit an unpleasant smell.

The public health impact of HnB products depends not only on whether they are less harmful than traditional cigarettes, but whether they encourage an increase or decrease in the prevalence of smoking. Findings from selected studies suggest that HnB tobacco products may create new nicotine addicted populations.

## Figures and Tables

**Figure 1 ijerph-17-00409-f001:**
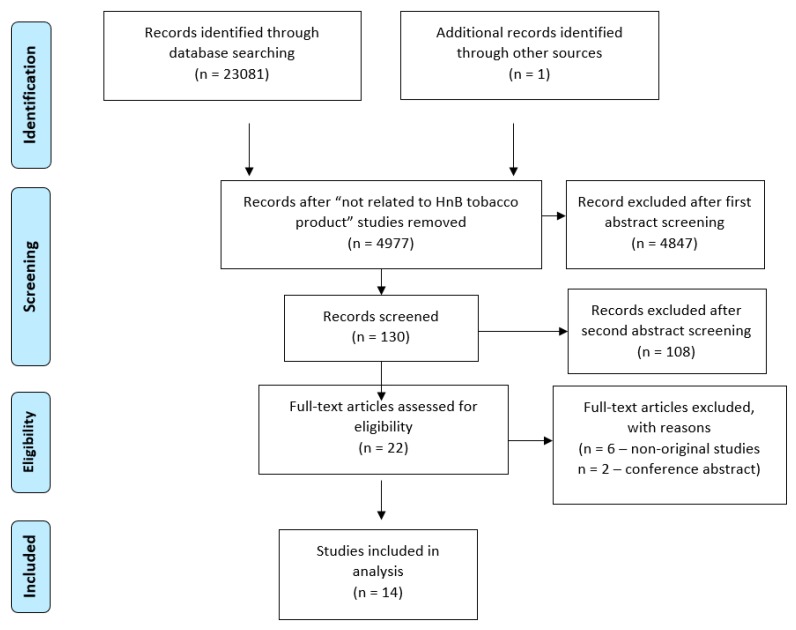
Public health issues.

**Table 1 ijerph-17-00409-t001:** Studies included in the review.

Authors, Year of Publication	Country	Funder	Study Design	Main Objective	Results	Number of Participants
Caputi T.L. et al., 2017 [12]	Japan	independent	Google search query data on HnB products	To estimate the scale and growth potential of HnBs	Queries for HnB products grew 100% between the products second (2016) and third years on the market.	5.9–7.5 million HnB-related Google searches in Japan
Czoli C. D. et al., 2018 [13]	Canada, England, USA	independent	Web-based cohort survey of people aged 16–19 years	To analyze youth awareness, interest in trying, and susceptibility to trying IQOS	7% of youth reported awareness of IQOS and 38.6% expressed interest in trying the product.	12,064
Hair E.C. et al., 2018 [14]	Switzerland, Japan	independent	Qualitative data via: (1) expert interviews, (2) semiotic analysis of IQOS packing and marketing materials, and (3) 12 focus groups with adults in Switzerland and Japan	To examine consumer perceptions and attitudes regarding HnBs and to document the product’s marketing strategies	IQOS packing and marketing analyses revealed the product is being marketed as a clean, chic and pure product, which appeals to young adults.	68
Kim J. et al., 2018 [24]	Korea	independent	Online survey of young adults aged 19–24 years	To analyze youth awareness, experience, and current use of HnB.	38.1% aware of IQOS, 5.7% HnB past users, and 3.5% current HnB users. All the current HnB users were triple users of traditional cigarettes and e-cigarettes.	228
Kioi Y. et al., 2018 [20]	Japan	independent	Online survey of adults aged 40–69 years	Analysis of the prevalence of electronic, HnB, and traditional cigarette use	HnB current or ever use was low (<0.1%).	4432
Lee A. et al., 2019 [21]	Korea	independent	Web-based anonymous self-administered survey of youth aged 12–18 years	Assessment of association of the HnB’s use with perceived stress, physical activity, and internet use	Significant associations between high perceived stress and cigarette use only, dual use of cigarette and e-cigarette, triple use of cigarette, e-cigarette, and HnB; not using the internet significantly increased the odds of use of all types of tobacco products	60,004
Lee P.N. et al., 2018 [25]	Japan	dependent	Computer simulations analyzing the impact on the population in different situations over a 20-year period from 1990	Estimation of the population health impact of introducing a reduced-risk tobacco product (RRP) into Japan	The introduction of an RRP into Japan will substantially reduce tobacco-related deaths.	irrelevant
Marynak K.L. et al., 2018 [15]	USA	independent	An internet survey of U.S. adults aged ≥18 years	Analysis of awareness and use of HnB tobacco products among U.S. adults	5.2% of U.S. adults aware of HnBs, including 9.9% of current cigarette smokers; 0.7% of U.S. adults, including 2.7% of current smokers, reported ever use of HnB; odds of ever use higher among current smokers than never smokers, and higher among adults aged <30 years than those aged ≥30 years	4107
Nyman, A.L., et al., 2018 [16]	USA	independent	Online survey of adults aged ≥18 years	Analysis of awareness and use of HnB tobacco products among U.S adults	A significant increase of HnB awareness among U.S. adults from 2016 to 2017 (from 0.5% to 1.1%). Men and adults under age 45 years had higher rates of awareness than women and those 45 and older. Non-white adults, cigarette smokers, and both current and former users of electronic nicotine delivery systems more likely to be using HnB	5992
Tabuchi, T., et al., 2018 [18]	Japan	independent	internet survey, respondents aged 15–69 years	To assess interest in HnB tobacco products (including IQOS, Ploom, and glo), its prevalence in 2015, 2016, and 2017, to examine the symptoms from exposure to secondhand HNB tobacco aerosol in Japan	Prevalence of IQOS users increased from 0.3% to 3.6% from 2015 to 2017. Rates of use of other HnBs remained low in 2017. Respondents who had seen the TV program more likely to have used IQOS. Half of never-smokers exposed to secondhand HnB aerosol reported at least one acute symptom, although these symptoms were not serious.	8240
Tabuchi, T., et al., 2016 [19]	Japan	independent	Web-based survey, respondents aged 15–69 years	To estimate awareness and use of e-cigarettes and heat-not-burn tobacco products among the Japanese population, including minors.	48% were aware of e-cigarettes and heat-not-burn tobacco products, 6.6% had ever used, 1.3% had used in the last 30 days. 7.8% and 8.4% of respondents used Ploom and iQOS, respectively, with a relatively higher percentage among the younger population.	8240
Wu Y.S. et al., 2019 [23]	Hong Kong, China	independent	telephone survey	To investigate HnB use and associated factors in Chinese adults in Hong Kong	11.3% of Hong Kong citizens were aware of HnBs and 1.0% had used it. Awareness was associated with aged 30–49 and higher socioeconomic status.	5131
Miyazaki Y. et al., 2018 [22]	Japan	independent	an Internet survey of Japanese adults aged 18–69	To analyze the relationship between educational attainment and e-cigarette and heat-not-burn tobacco use.	No straightforward associations between educational attainment and e-cigarette ever-use or heat-not-burn tobacco use. An inverse association between educational attainment and combustible cigarette smoking.	7338
Brose L.S. et al., 2018 [17]	Great Britain	independent	An online survey of Great Britain respondents aged ≥17 years	To estimate awareness and use of HnB tobacco products in Great Britain	9.3% of respondents reported awareness; this included 0.9% who had tried or used the products in the past and 0.8% currently using	12,696

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
