# Peer review of "Heat Not Burn Tobacco Product—A New Global Trend: Impact of Heat-Not-Burn Tobacco Products on Public Health, a Systematic Review"

_ijerph, 2020, doi:10.3390/ijerph17020409_

Round 1
Reviewer 1 Report
REVIEW OF Int. J. Environ. Res. Public Health 2019, 16, x; doi: FOR PEER REVIEW
Subject manuscript will require much revision before it is acceptable for publication.
First, it appears that the authors wanted to get a paper published without doing serious research to support their apparent claim the attractiveness of HnB is improper due to their purported danger to the users. The attractiveness of HnB is only an issue if the products pose a significant risks of adverse health risks to the users and/or bystanders. The authors have failed to support the notion that HnB products pose a significant risks of adverse health risks to the users and/or bystanders
The authors incompletely cite Aokage et al. in their opening paragraph. This reviewer uses the term “incompletely” because later on in that journal article, Aokage mentions examples of acute eosinophilic pneumonia (AEP) that were caused by conventional tobacco products. If the authors wish to discuss a possible relationship between HnB and AEP, they should also cite inter alia Kamada et al., Respirology Case Reports, 4 (6), 2016, e00190, who wrote, “Acute eosinophilic pneumonia (AEP) is a rare disorder characterized by hypoxemia, pulmonary infiltrates, and pulmonary eosinophilia. AEP occurs secondary to drug exposure or hypersensitivity reactions to an inhaled antigen. Tobacco smoking is one of the main causes of AEP.” Moreover, several other relevant journal articles have been published since the authors submitted their manuscript.
If they were truly concerned about HnB products, particularly the IQOS, they would have sampled product from the various markets, determined the range of human puffing behaviors, and ran smoke chemical and in vitro toxicological tests on the aerosols generated under human puffing conditions that yielded the most toxic smoke. The authors lamented the lack on non-industry studies on the IQOS. They had their chance to do something about the lack of studies, and they failed to do so. Numerous alcoholic beverage and food with potential adverse effects are purchased by consumers because of their attractiveness. Attractiveness alone is not a reason to denigrate a product, nor is its nicotine content. Even the US FDA recognized that in giving MRTP status to certain brand styles of Swedish Snus.
Second, the authors appear to have neglected much of the research on HnB because it was done by scientists employed by the tobacco industry. Does that automatically make it so bad that authors can in good conscious fail to cite it? Perhaps that authors were not taught to cite ALL RELEVANT RESEARCH. Just because you cite research, doesn't mean that you accept all the findings, but it does mean you read it thoroughly, you understand the experimental work, you decide if it was valid or not in whole or part, and you explain to the readership why you determined that the research was erroneous, if that were your conclusion. If the authors submit a revised manuscript, it should include references to all HnB literature and the authors' rationale for determining whether or not it indicates potential toxicological issues with HnB use. The authors should seriously consider consulting a board-certified toxicologist with expertise in the chemistry and toxicology of conventional and HnB products. Just because there are increases in some analytes versus a conventional combustible cigarette, does not show that the smoke is not less hazardous. In particular, the authors should consult the following journal article: Ito et al., An inter-laboratory in vitro assessment of cigarettes and next generation nicotine delivery products, Toxicology Letters 315 (2019) 14–22, and the references cited therein.
Reviewer 2 Report
As a reviewer I have the following remarks
Line 40, I suggest put (COPD) here as later (line 75) you use again. Lines 151-154, three countries are listed and later “in this region”. Line 180: “this year” – I suggest to specify 2019, as in 2, 3 years we have many options.Thank you.
Reviewer 3 Report
I am honored to review this manuscript written by Dr Alekasandra Ratajczk. This manuscript indicates that HnB cigarette has more interest in young adult population than middle old, some adverse effect exits but that is less than traditional tobacco.
I would like ask the author a couple of minor revision and give my comment to the author.
Minor comments.
(1) "7.0% and 38,6% respectively" at line 128 >> "7.0% and 38.6% respectively" ??
(2) "in his review" at line 256 >> " in this review" ??
(3) At line 342 to 345 in Reference, there are two articles indicated in No.22. Which sentence refer the article written by Dr Brose, L et al (Tobacco Regulatory Science 2018 4:44-50) ?
My comment:
The author mentions "Present studies suggest that HnB tobacco products have the potential to be a reduced risk product for public health" at line 266 and in Abstract. I think there are not enough evidences to justify it.
Only the reference No.21 mentions indirectly the potential effect on the chronic diseases which traditionally linked to cigar habit (CVDs, cancer, COPD e.g.), however, there are some case reports where HnB tobacco causes acute pneumonia and respiratory failure. It may be worthy to note.
Takahiro Kamada, et al. "Acute eosinophilic pneumonia following heat‐not‐burn cigarette smoking." Respirology Case Reports, Vol.4, Issue6, 2016
Toshiyuki Aokage, et al. "Heat-not-burn cigarettes induce fulminant acute eosinophilic pneumonia requiring extracorporeal membrane oxygenation." Respiratory Medicine Case Reports, Vol.26, 87-90, 2019
Ghinai I et al. "E-cigarette Product Use, or Vaping, Among Persons with Associated Lung Injury - Illinois and Wisconsin, April-September 2019." MMWR Morb Mortal Wkly Rep 2019;68:865-869.
Round 2
Reviewer 1 Report
REVIEW OF Int. J. Environ. Res. Public Health 2019, 16, x; doi: Version 2
The authors are to be commended for their responses to the critique from the reviewers.
This reviewer apologizes for not adding the following to his initial review:
The basis for the authors' manuscript appears to be that HnB products present an unreasonable risk to the users of said products and that marketing of such projects is not appropriate. However, at least one HnB, the IQOS intended for sale in the US received approval from the US FDA. In its press release (https://www.fda.gov/news-events/press-announcements/fda-permits-sale-iqos-tobacco-heating-system-through-premarket-tobacco-product-application-pathway), the FDA wrote:
Following a rigorous science-based review through the premarket tobacco product application (PMTA) pathway, the agency determined that authorizing these products for the U.S. market is appropriate for the protection of the public health because, among several key considerations, the products produce fewer or lower levels of some toxins than combustible cigarettes. The products authorized for sale include the IQOS device, Marlboro Heatsticks, Marlboro Smooth Menthol Heatsticks and Marlboro Fresh Menthol Heatsticks. While today’s action permits the tobacco products to be sold in the U.S., it does not mean these products are safe or “FDA approved.” All tobacco products are potentially harmful and addictive and those who do not use tobacco products should continue not to. Additionally, today’s action is not a decision on the separate modified risk tobacco product (MRTP) applications that the company also submitted for these products to market them with claims of reduced exposure or reduced risk.
Since much of the authors' manuscript focuses on the IQOS, the above passage should be added and referenced. I suggest adding it at Line 35, before the existing text. The authors are then free to add the current text beginning at Line 35 along with other negative information about the IQOS and other HnB. In this context the authors should consider the following and amend their manuscript accordingly.
Some HnB devices reportedly expose the tobacco to lower temperatures than in the IQOS and may emit less HPHC. Are there any chemical and/or toxicological data relevant to differences in adverse health effects among the IQOS and other HnB?
Not much has been said about the ETS (mouthspill and exhaled mainstream smoke) from HnB. Are there potentially adverse health effects to bystanders who might inhale the ETS? For example, if use of HnB products, and the IQOS, in particular, presents no chronic adverse effects on the users, and there is no adverse effects (including noxious odors) to bystanders, and there is no third-hand smoke, then there is no basis for the authors' manuscript. Marketing of such products would be analogous to marketing of alcoholic beverages and food products known to cause adverse health effects (for example, highly-spiced snacks such as Cheetos Flamin' Hot Crunchy Cheese Flavored Snacks).
It is tempting for many researchers and regulators to compare quantitative lists of mainstream smoke deliveries from the IQOS and other HnB products with those from conventional combustible cigarette products whether they be reference products such as the 3R4F reference cigarette or leading brand-styles of commercial cigarette products. An example of such comparisons can be found in the paper by St. Helen et al. [Tob Control 2018;27:s30–s36 (the authors should cite that journal article)]. One of the main problems with such lists is that the toxicological properties of any given constituent may be different when the constituent is assayed alone as opposed to being in a complex mixture. Another problem with such lists is that it takes substantial experience to determine whether a given constituent is the result of a deliberately added flavor or humectant or is a pyrolysis product from the tobacco or other components of the device. Most would agree that carbon monoxide (CO) is a constituent whose toxicological properties are not effected by other smoke components. Indeed, some experts on the toxicological properties of cigarette smoke believe that CO is the cause of many smoking related diseases.
Another class of smoke toxicants are the free radicals. Most laboratories do not have the instrumentation and expertise to determine the levels and structures of the free radicals in mainstream cigarette smoke or the mainstream aerosol from HnB devices. Unfortunately the difficulty of determining the nature and concentrations of such reactive species has kept them off the various lists of smoke toxicants promulgated by various regulatory bodies. Moreover, one of the hypotheses put forward by PMI scientists in their presentation to the US FDA TPSAC was the disease-causing potential of free radicals residing on the “soot” particles that make up the majority of the the nonvolatile phase of the mainstream cigarette smoke aerosol. No soot therefore very low levels of free radicals, much less smoking-related diseases.
Since the concentrations of CO and free radicals in the mainstream aerosols from the IQOS and other HnB products are much reduced versus their concentrations in the mainstream smoker from conventional combustible cigarettes, any reliance on lists of other smoke constituents (e.g., US FDA HPHC lists) is suspect. The gold standard here is biological markers of dose and harm developed from human clinical studies. Due to the expense and difficulties of such clinical studies various in vitro and in vivo toxicological assays are often used, particularly for ascertaining which prototypes would likely show toxicity in human clinical studies.
Now the conundrum for the authors as well as the regulators is that human smoking of the IQOS and other HnB in clinical studies will show increased levels of biological makers of dose and/or harm relative to said markers in the nonsmoking control group. For example, let us assume that exclusive use of any given HnB is projected to reduce the incidence of serious smoking-related diseases (heart disease, COPD, respiratory-tract cancers) by 80% and adverse effects to bystanders by 95%. The question the authors should address is, ”Is there sufficient public health benefits for such devices be marketed to adults in the same manner as beer and other alcoholic beverages?”
In answering that question the authors need to update their list of citations to include recent toxicological studies to include inter alia, Davis et al. [Toxicology in Vitro 61 (2019) 104652], and give their professional opinion as to the value of such studies in determining the potential of the IQOS and other HnB to cause adverse health effects in adult users of those products.
In addition to the above, this reviewer has the following suggestions for a revised manuscript.
There are literature references for all the products mentioned in Lines 49-52. Indeed, with respect to Premier, the authors should cite and read, “Chemical and Biological Studies on New Cigarette Prototypes that Heat Instead of Burn Tobacco”. The authors can download a copy of the book at https://www.industrydocuments.ucsf.edu/tobacco/docs/#id=pyhd0140.
Another must read and cite is A Safer Cigarette? A Comparative Study. A Consensus Report, ECLIPSE Expert Panel, Inhal Toxicol. 2000;12 Suppl 5:1-58. doi: 10.1080/08958378.2000.11720735.
There are numerous journal articles dealing with ACCORD and other electrically heated smoking devices as well as PREMIER and ECLIPSE. The authors, in their determination of the potential adverse health effects of IQOS and other HnB, should compare contemporary clinical data with the clinical data obtained on Premier, Eclipse, and Accord.
At Lines 138-152, the authors comment on two studies on U.S. adults, particularly with respect to younger adults. While there was heavy advertising of Premier and then Eclipse from 1988 to 2005, there was little advertising after that. There was little advertising of ACCORD. Therefore, would the responses obtained from the adults surveyed have been different if they had not been prompted with descriptions of HnB? Also, please note that the first introduction of the US version of the IQOS is just beginning. Moreover, it has been common in the US for the younger adults with more financial resources and technological savvy to purchase and use new technology than do older adults. Is this a good thing or a bad thing?
In conclusion, for the authors' manuscript to be acceptable for publication and be valuable for the public health community, they must use all available information to determine the public health benefits of marketing of the IQOS and similar HnB devices. Their approach to this topic must be balanced. If there are flaws in the IQOS and other HnB that would require restricted marketing to adult consumers, then the authors should point them out.
Round 3
Reviewer 1 Report
By this time, I am sure that the authors are tired of getting my critique of their revised manuscripts. However, the authors have a unique topic; and their manuscript needs to be above reproach. My job as a reviewer is not only to remove dubious text from their manuscript, but also offer suggestions to make it better.
First, I what to reconfirm with the authors their main hypothesis: There are sufficient toxicological concerns with the current HnB products (including the IQOS) to warrant restrictions of the sales and marketing of such products. Am I correct? If not, the authors need to rewrite the appropriate sections of their manuscript so that their hypothesis is clearly stated.
The authors should up date their references to include iqos marketing and heat-not-burn marketing. I just checked those search terms in PubMed and there are about six new articles that the authors should reference.
Second, I want to alert the authors to the following scenario:
It is likely the makers current HnB products are working to develop products that do not have the potential for adverse health effects that the current products have. Supposing there are HnB products that have less than a 10% chance of causing adverse health effects and those that might occur would be entirely reversible if the user of said HnB product discontinued use. Moreover, said products would cause no adverse health effects to by standers. No CO, no formaldehyde, no TSNAs, no irritating agents. You could be in the proverbial, “smoke-filled room” without discomfort or irritation to the eyes and respiratory tract.
How would the authors feel about the sales and marketing of such products? They should seriously consider such an eventuality. Sales of conventional cigarettes are way down in many markets. There is public sentiment, regulatory actions, and legislative actions against the sales of e-vapor products. Unless that changes, HnB may be the only way forward for the major cigarette companies.
The authors discussion of HPHC needs to include the following citations:
Stephens WE. Tob Control 2018;27:10–17. (http://dx.doi.org/10.1136/ tobaccocontrol-2017-053808).
Mallock N, et al. Front Public Health. 2019 Oct 10;7:287. doi: 10.3389/fpubh.2019.00287. eCollection 2019.
Gasparyan H, et al. Regulatory Toxicology and Pharmacology 99 (2018) 131–141 (doi: 10.1016/j.yrtph.2018.09.016).
All of the above articles are open access and can be downloaded without charge.
The authors may also want to obtain (not open access) and cite, the following article:
Kopa PN, Pawliczak R. IQOS - a heat-not-burn (HnB) tobacco product - chemical composition and possible impact on oxidative stress and inflammatory response. A systematic review. Toxicol Mech Methods. 2019 Oct 2:1-7. doi: 10.1080/15376516.2019.1669245.
The authors should also consider citing (open access): Munakata et al., Oxidative stress responses in human bronchial epithelial cells exposed to cigarette smoke and vapor from tobacco- and nicotine-containing products. Regul Toxicol Pharmacol. 2018 Nov;99:122-128. doi: 10.1016/j.yrtph.2018.09.009.
This reviewer suggests Munakata because in vitro assays may provide a better estimation of adverse health effects that do quantitative lists of HPHC.
Crooks et al., Evaluation of flavourings potentially used in a heated tobacco product: Chemical analysis, in vitro mutagenicity, genotoxicity, cytotoxicity and in vitro tumour promoting activity. Food and Chemical Toxicology 118 (2018) 940–952 (https://doi.org/10.1016/j.fct.2018.05.058) provides a comparison of smoke chemical data (HPHC) with toxicological data from three types of in vitro assays.
In terms of marketing of the IQOS in the USA (known as Marlboro Heat Sticks), the authors should view the video at https://www.nbc12.com/2019/11/14/philip-morris-launching-cigarette-alternative-richmond/. Based on other information, one cannot get into a similar store in Atlanta, GA, unless one can prove that customer is age 21 or over AND is a current smoker.
This reviewer does not have a journal citation in the above. Another website with useful information is https://countertobacco.org/emerging-product-watch-iqos-introduced-in-atlanta-and-richmond/.
There is an article in the Atlanta Journal Constitution at https://countertobacco.org/emerging-product-watch-iqos-introduced-in-atlanta-and-richmond/
The authors should add at least one of these citations to their manuscript when discussing IQOS and HnB in the USA.
